# Impacts of Aging Agricultural Labor Force on Land Transfer: An Empirical Analysis Based on the China Family Panel Studies

Chaozhu Li [1], Xiaoliang Li [2,*], Jiaxu Wang [3,*] and Tianchu Feng [2]

1    China Institute for Rural Studies, Tsinghua University, Beijing 100084, China
2    Jiyang College, Zhejiang Agriculture and Forestry University, Zhuji 311800, China
3    China Electronics Standardization Institute, Beijing 100010, China
*    Correspondence: lixiaoliang@zafu.edu.cn (X.L.); wangjx1@cesi.cn (J.W.)

**Abstract:** Aging is an important trend in the global demographic, with rural population aging becoming a significant challenge due to its faster pace and profound implications. Although the most significant impact of the aging agricultural labor force occurs in agricultural production and land use, little is known about their relationship. Based on the 2010–2018 data from the Chinese Family Panel Studies, this study uses the panel probit model to analyze the impact of the aging agricultural labor force on land transfer and tests its influence mechanism from the aspects of health effect and pension insurance effect. The results show the following: (1) there is an inverted U-shaped relationship between the aging of the agricultural labor force and land rent-in—with the deepening of the aging of the agricultural labor force, the aging of the agricultural labor force shifts from promoting land rent-in to inhibiting it; (2) there is a U-shaped relationship between the aging of the agricultural labor force and land rent-out—with the deepening of the aging of the agricultural labor force, the aging of the agricultural labor force shifts from inhibiting land rent-out to promoting it; and (3) the impact of the aging agricultural labor force on land rent-in and rent-out is significantly affected by farmers' health levels, however pension insurance does not play a significant moderating role. Therefore, the government should strengthen the land transfer market and improve the level of pension insurance for the elderly rural agricultural labor force.

**Keywords:** aging; land rent-in; land rent-out; health level; pension insurance

## 1. Introduction

Population aging is a global problem [1,2]. According to the 2019 Revision of World Population Prospects released by the United Nations, it is estimated that, by 2050, the proportion of people over 65 years old in the world will rise from 11% to 16% [3], while rural population aging will face greater challenges due to faster and deeper development. Although China entered the aging stage later than developed countries, the aging population in China is huge, and the aging trend is extremely rapid, especially in rural areas [4]. According to the seventh national census data in 2021, 17.72% of people over 65 years are in rural China, which is 6.61% higher than that in urban areas. "Elderly agriculture" has become a typical characteristic of Chinese agriculture [5].

Existing studies analyze the factors of land transfer from the perspective of property rights [6–9], non-agricultural employment [10–13], household characteristics [14,15], and land endowment [16,17]. For example, Deininger and Feder [18] argue that a clear and secure rural land property rights system is an important prerequisite for the establishment of a land transfer market, and that the confirmation of contracted land rights can help mobilize farmers to participate in the land transfer market and promote land transfer. Su et al. [19] concluded that non-agricultural employment has a significant positive impact on farmers' land rent-out, while it has no significant impact on farmers' land rent-in. Yu et al. [15] found that male heads of households with higher education levels

were more inclined to rent-in land. Xiao et al. [20] confirmed that the higher the degree of land fragmentation, the more willing farmers are to rent-out land. With the out-migration of the young rural labor force and a change in family population structure, some scholars have begun to focus on the impact of aging on land transfer [21,22]; however, the research conclusions are not consistent. Liu et al. [23] believe that, when the supply of young labor in farmer households is reduced to a certain extent and the remaining elderly labor force cannot manage all the land, they will rent-out land to obtain rental income. However, Yang and Chen [24] assume that, if the degree of aging is relatively light, the labor cost is low, or there is the replacement of agricultural machinery [15], aging will not have a significant impact on land rent-out.

Previous studies have helped us to gain a better understanding of the relationship between aging and land transfer [23,25–27], but further research is still needed. First, most of the literature used cross-sectional data and analyzed the impacts of aging on land rent-out, while fewer studies used large-sample panel data to analyze the impact of aging agricultural labor on land rent-in and rent-out. Second, existing studies only considered the linear impact of aging on the relationship between land transfer, while few studies analyzed the possible non-linear relationship between the aging agricultural labor force and land transfer. Third, existing studies focused on the direct impact of aging on land transfer; however, its mechanisms have not been explored. Thus, the main purpose of this study is to analyze the impact and mechanism of aging on land transfer. The marginal contribution of this paper is that we use a large sample of panel data to analyze the non-linear relationship of the aging agricultural labor force on land transfer and test the moderating role of health level and pension insurance to answer the mechanism of the effect of the aging agricultural labor force on land transfer, which is important for stabilizing agricultural production and ensuring food security.

## 2. Theoretical Analysis and Research Hypothesis

Based on the basic model of Carter and Yao [28], consider a representative farmer $i$ with two initial endowments, labor endowments $L_{0,i}$ and land endowments $T_{0,i}$, as well as a certain agricultural production capacity $\alpha_i$. To maximize household income, farmers need to allocate labor and land. Farmers can rent-in or rent-out land, and they can hire labor or go out for work [29]. $L'$ represents the actual labor force engaged in agricultural production, $L_{a,i}$ denotes the household's own labor force engaged in agricultural production, $L_{n,i}$ indicates the labor force in non-farm employment, and $L_e$ denotes the employed agricultural labor. $T_i$ denotes the actual land cultivated by farmers and $T_{r,i}$ denotes the land transfer area of farmers. If $T_{r,i} = T_i - T_{0,i} > 0$, it means that the farmer rents in land. If $T_{r,i} = T_i - T_{0,i} < 0$, it means that the farmer rents out land. Assuming that there is no supervision cost in hired labor and both sides of the transfer transaction have the same transaction cost $c$, the non-farm employment wage rate is $\omega_1$, and the wage rate of hired labor is $\omega_2$. To simplify the analysis, normalize the price of agricultural products to 1. Thus, the optimization problem faced by farmer $i$ can be formulated as

$$Max \ \alpha_i f(L', T_i) + \omega_1 L_{n,i} - \omega_2 L_e - I^{in}[(T_i - T_{0,i})(r + c)] + I^{out}[(T_{0,i} - T_i)(r - c)] \quad (1)$$

In Equation (1), $r$ is the land rental rate, and $I^{in}$ and $I^{out}$ represent whether the farmers rent-in and rent-out land.

$$Q = \alpha_i f(L', T_i) = \alpha_i T_i^{\beta} L'^{1-\beta} \quad (2)$$

$Q$ is the agricultural production function of farmer $i$, satisfying the following assumptions (subscripts $i$ are omitted in the following derivations for ease of presentation): $f_{L'} > 0$, $f_T > 0$, $f_{L'L'} < 0$, $f_{TT} < 0$, $f_{L'T} > 0$ and $f_{L'L'}f_{TT} - f_{L'T} > 0$.

Solving the optimization problem yields the following first-order conditions.

$$\alpha f_{L_a}(L', T) = \omega_2 - (\omega_2 - \omega_1)\frac{\partial L_a}{\partial L'} \quad (3)$$

$$\alpha f_T(L', T) = r + c \text{ (rent-in)} \tag{4}$$

$$\alpha f_T(L', T) = r - c \text{ (rent-out)} \tag{5}$$

In Equation (3), the allocation of labor resources by the farm household depends on the marginal income $\alpha f_T(L', T)$ received from agricultural production compared with $\omega_1$, $\omega_2$, and $\frac{\partial L_a}{\partial L'}$, where $\frac{\partial L_a}{\partial L'}$ is greater than 0 and less than 1. Equations (4) to Equation (5) show that the allocation of land resources depends on the marginal output of land $\alpha f_T(L', T)$ compared to the level of net land rent $(r \pm c)$. If the marginal output of the farmer's land is greater than the level of net land rent, the farmer will choose to rent-in land. Conversely, the farmer will choose to rent-out land. For farmers who do not transfer land, the decision conditions meet:

$$r - c < \alpha f_T(L', T) < r + c \tag{6}$$

By dividing both sides of Equation (6) by $f_T(L', T)$ at the same time, we obtain two boundary points $\alpha_l = (r - c)f_T^{-1}(L', T)$ and $\alpha_u = (r + c)f_T^{-1}(L', T)$ of farmers' productive capacity. In order to obtain the influence of changes in agricultural production capacity $\alpha$ on the amount of land rent-in (out) by farmers, we fully differentiate $\alpha$ on both sides of Equation (3), Equation (4), or Equation (5), and, according to Gramer's rule, it follows that

$$\frac{\partial T_r}{\partial a} = \frac{f_{TL'}f_{L'} - f_T f_{L'L'}}{f_{TT}f_{L'L'} - (f_{TL'})^2} > 0 \tag{7}$$

$$\frac{\partial T_r}{\partial a} = -\frac{f_{TL'}f_{L'} - f_T f_{L'L'}}{f_{TT}f_{L'L'} - (f_{TL'})^2} < 0 \tag{8}$$

Equations (7) and (8) show that farmers' land rent-in (rent-out) strictly increases with their high (low) productive capacity $\alpha$. This indicates that the higher the agricultural production capacity of farmers, the larger the land scale that farmers are willing and able to operate, and the more inclined they are to rent-in land rather than rent-out land. Conversely, the lower the agricultural production capacity of farmers, the smaller the land scale they are willing and able to operate, and the more inclined they are to rent-out land rather than rent-in land.

Aging affects farmers' agricultural production capacity [27] and thus affects the land transfer. The influence of aging on farmers' land transfer is not linear. First, when aging is lower than a certain level, and farmers still have agricultural production capacity, in order to obtain economic benefits, they may rent-in land and not rent-out land [24]. Second, when aging exceeds a certain level, and farmers cannot manage all the land, the impact of aging on land transfer is influenced by the rural labor market. Assuming that there is no developed labor market in rural areas and farmers can only rely on family labor force to engage in agricultural production, when aging exceeds a certain degree, aging will promote farmers to rent-out land, but inhibit farmers to rent-in land [23,30]. Assuming that there is a developed labor market in rural areas, when aging deepens, farmers can alleviate the shortage of agricultural labor force by employing labor. When farmers' income from agricultural production by employing a labor force is higher than their income from renting land, they will maintain or expand the current land scale by employing a labor force. However, when farmers' income from agricultural production by employing a labor force is lower than that from renting out land, rational farmers will not choose to employ a labor force for agricultural production, but will rent-out land to obtain rental income [25]. However, at present, China's rural labor market faces the dual pressure of insufficient supply and high labor costs [31]. First, China's rural areas are facing the problem of aging, with a decrease in help among farmers and the decrease in labor supply in the agricultural labor market. Elderly farmers may find it difficult to cope with the problem of reduced labor input by helping or hiring workers [32]. Second, in 2021, the urbanization rate of China's permanent resident population reached 64.72%, and a large number of young and middle-aged rural labor force rushed to cities and non-agricultural industries, which changed the

labor supply at a given price level, thus raising the rural labor price [24]. Moreover, the rate of increase is much faster than that of the per capita net income of rural residents in the same period [33], thus pushing the cost of employment agriculture to break through the critical point of land transfer [26]. Zou et al. [26] found that, when farmers are too old to cultivate their land, 45.45% of them would rent their land, while only 7.47% would hire agricultural labor. Therefore, Hypothesis 1 is proposed:

**H1.** *The aging of agricultural labor force has an inverted U-shaped impact on land rent-in and a positive U-shaped impact on land rent-out.*

The impact of the aging agricultural labor force on land transfer is affected by health level and pension insurance. With the aging of the elderly labor force, their physiological function will continue to decline, and their physical health status will gradually deteriorate, reducing their ability to bear high-intensity agricultural production activities; hence, it will become difficult to meet the needs of modern agricultural production [34]. Therefore, compared with farmers with low health levels, farmers with high health levels can alleviate the negative impact of the aging agricultural labor force on land rent-in and inhibit the positive impact on land rent-out. For a long time, family support has been a traditional form of rural pension security, and land plays the role of social security, providing life security, pension security, and employment security [35]. If farmers do not have pension insurance, elderly farmers may hold or rent-in land to ensure their livelihood rather than rent-out land. For rural households with pension insurance, stable pension income can meet the necessary monetary expenditure needs of elderly farmers, and pension insurance can replace the function of land endowment security, inhibiting farmers from renting in land, but encouraging farmers to rent-out land [36]. Therefore, compared with farmers without pension insurance, farmers with pension insurance can strengthen the negative impact of the aging of the agricultural labor force on land rent-in and the positive impact on land rent-out. Hypothesis 2 and Hypothesis 3 are proposed:

**H2.** *The agricultural labor force's health level restricts aging's impact on rural households' land transfer. Compared with those with low health levels, the agricultural labor force with higher health levels can alleviate the negative impact of the aging agricultural labor force on land rent-in and inhibit the positive impact on land rent-out.*

**H3.** *The influence of aging on rural households' land transfer is constrained by pension insurance. Compared with those without pension insurance, rural households with pension insurance can strengthen the negative impact of the aging agricultural labor force on land rent-in and the positive impact on land rent-out.*

## 3. Methodology

### 3.1. Data Source

The data used in this study are from the CFPS, conducted by the China Social Science Survey Center of Peking University. The survey was officially conducted in 2010, and the survey sample covered 25 provinces in China, accounting for 95% of the total population of mainland China [37]. In 2012, 2014, 2016, and 2018, the sample households were followed up to collect data from three levels: community, household, and individual, reflecting the social, economic, demographic, and health changes in China in different periods, which is highly representative. Since the research object is the aging of the agricultural labor force, the data are processed as follows: First, according to the types of urban and rural areas of the National Bureau of Statistics, the sample of households living in rural areas is retained, and the sample of urban areas is excluded. Second, the individual, household, and community databases are merged year by year to retain farmers who own land. Third, samples with missing key information, such as personal and household characteristics,

are removed. Finally, 26,909 valid samples are obtained, of which 6980, 5738, 5864, 4681, and 3646[1] are for 2010, 2012, 2014, 2016, and 2018, respectively. The distribution of sample provinces is shown in Figure 1.

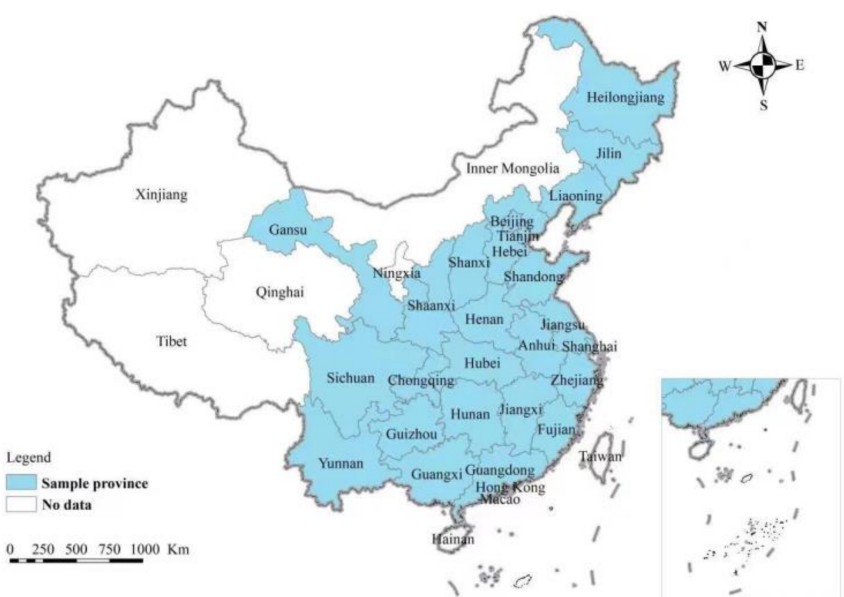

**Figure 1.** The spatial distribution of sample provinces.

Table 1 describes the CFPS survey sample from 2010 to 2018. At the household level, the average age of the agricultural labor force is 50.44 years old, showing an increasing trend. These households where at least one member's age is over 60 account for 44.15% of the total sample, and the proportion increases annually. The 2018 survey shows that more than half of the households have an elderly member over 60 years old, and nearly 18% of the households consist entirely of elderly members over 60 years old. The scale of contracted land per household is 11.40 mu, and the total land transfer rate is 25.77%, in which the rent-in land rate is 15.32%, and the rent-out land rate is 10.45%. In different years, the rent-in land rate is relatively stable, while the rent-out land rate increases annually.

**Table 1.** Basic information of CFPS survey samples from 2010 to 2018.

| Variables | 2010 | 2012 | 2014 | 2016 | 2018 | Total |
|---|---|---|---|---|---|---|
| Number of households | 6980 | 5738 | 5864 | 4681 | 3646 | 26,909 |
| Average age of agricultural labor force | 47.58 | 47.84 | 51.19 | 53.16 | 54.97 | 50.44 |
| At least one elderly person in the household (%) | 35.00 | 39.32 | 48.47 | 52.36 | 51.81 | 44.15 |
| The labor force in the household is all elderly (%) | 11.78 | 13.99 | 15.16 | 18.01 | 17.96 | 15.01 |
| Per household contracted land area (mu [a]) | 10.96 | 10.60 | 11.56 | 12.03 | 12.40 | 11.40 |
| Land rent-in rate (%) | 16.00 | 15.33 | 15.76 | 16.04 | 12.37 | 15.32 |
| Land rent-out rate (%) | 4.23 | 10.98 | 11.19 | 13.74 | 16.10 | 10.45 |

Note: [a] 1 mu = 667 m$^2$ or 0.067 ha.

Figure 2 depicts the land transfer of farmers in six different age groups based on the average age of the agricultural labor force. It can be found that the different age groups of farmers have significant differences in land rent-in and rent-out rates. First, the land rent-in rate of farmers under 60 years old is significantly higher than the land rent-out rate, while the land rent-out rate of farmers over 60 years old is significantly higher than the land rent-in rate. Second, except for the age group under 30 years old, with the increase in the average age of the agricultural labor force, the land rent-in rate decreases gradually while the land rent-out rate keeps increasing.

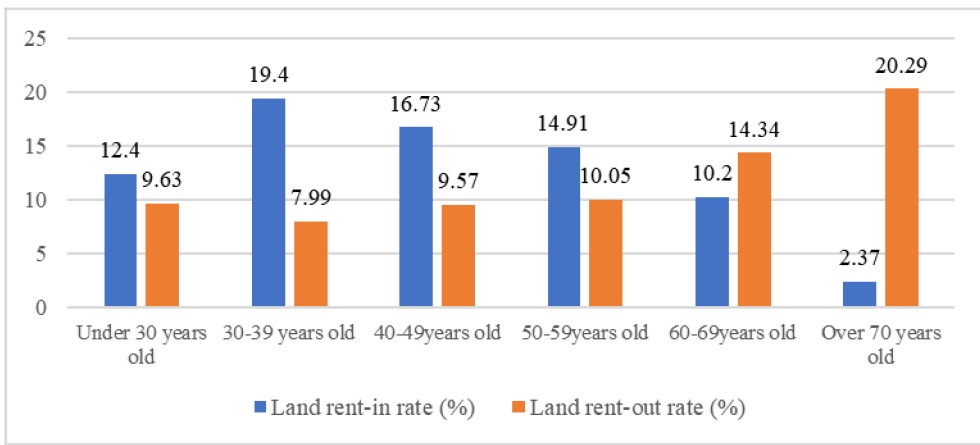

**Figure 2.** Land transfer of sample farmers by age of agricultural labor force group.

*3.2. Variable Selection*

### 3.2.1. Dependent Variables

The dependent variable of this study is farmers' land transfer. Farmers' land rent-in is measured according to "whether to rent other farmers' land". If farmers rent-in land, the value is 1. Otherwise, the value is 0. Farmers' land rent-out is measured according to "whether to rent land to others". If farmers rent-out land, the value is 1. Otherwise, the value is 0.

### 3.2.2. Independent Variable

The independent variable is the aging of the agricultural labor force. There are three ways in which aging is currently measured in academia. One is to use the age of the household head or the average age of the labor force as a proxy for aging [38,39]. The second is to divide farmers into aged households and non-aged households based on a certain age [26]. The third is to use the proportion of elderly population in a family or the proportion of labor force in the total population of a family as a measurement index of aging [27]. Considering that China's agricultural production is mainly based on the farm household as the production and management unit, and the division of labor among household members is a process of optimization according to the characteristics of labor endowment, in this study, the average age of the agricultural labor force is used to measure the aging of the agricultural labor force [39]. Meanwhile, the above indicators were used to replace the average age of the agricultural labor force in the robustness test.

### 3.2.3. Moderating Variables

The two moderating variables in this study are the health level of the agricultural labor force and whether there is pension insurance. The health level of the agricultural labor force is a binary dummy variable. A high average health level of the agricultural labor force takes the value of 1 and a low average health level takes the value of 0[2]. Pension insurance is a binary dummy variable. If a family member has pension insurance, the value is 1; otherwise, the value is 0.

### 3.2.4. Control Variables

Referring to relevant studies [40–45], we control for household characteristics and village characteristics. Household characteristics variables include the average years of education of the agricultural labor force, net agricultural income, whether to hire workers, the number of the labor force, and per capita land area. Village characteristics variables include village topography and whether the village has access roads. Moreover, some unobservable factors that change with time and region may constitute the endogenous problem of the econometric model; thus, dummy variables for time and region are controlled. The definitions and descriptive statistics of the variables are shown in Table 2.

**Table 2.** Variables and descriptive statistics.

| Variables | Definition | Mean | Std.Err. |
|---|---|---|---|
| Inrent | Whether farmers rent-in land (1 = yes; 0 = no) | 0.153 | 0.360 |
| Outrent | Whether farmers rent-out land (1 = yes; 0 = no) | 0.105 | 0.306 |
| Aging | Average age of the agricultural labor force (years) | 50.434 | 11.066 |
| Aging * Aging | The square term of the average age of the agricultural labor force | 2666.037 | 1150.654 |
| Health | Whether the average health of the agricultural labor force is high (high = 1; low = 0) | 0.628 | 0.483 |
| Pension | Whether family members have pension insurance (yes = 1; no = 0) | 0.310 | 0.463 |
| Eduy | Average years of education of the agricultural labor force | 5.502 | 3.370 |
| Net_agri | Net agricultural income (10,000 CNY) | 0.651 | 1.745 |
| Hire | Whether to hire workers (1 = yes; 0 = no) | 0.146 | 0.353 |
| Labor | Total labor force in the family (person) | 2.782 | 1.274 |
| Land_per | Per capita land area of family (mu/person) | 3.241 | 7.921 |
| Landscape | The landform of the village is plain = 1, otherwise = 0 | 0.395 | 0.489 |
| Road | Whether the village has access to roads (yes = 1; no = 0) | 0.881 | 0.362 |
| Dum_area | Regional dummy variable (east, middle, west, northeast) | NA | NA |
| Dum_year | Time dummy variable (2010/2012/2014/2016/2018) | NA | NA |

### 3.3. Model Selection

The data sample selected for this study is panel data, and since the dependent variables are discrete, the model selection requires a discrete panel data model. There are two types of discrete panel data models: one is the panel probit model and the other is the panel Logit model. Although the probit model requires the random error term to obey a normal distribution, the Logit model has no such requirement. Referring to Liu and Liu [46], a panel Logit model is used to analyze the impact of the aging agricultural labor force on land transfer as follows:

$$Inrent_{it} = \alpha_1 + \beta_1 Aging_{it} + \beta_2 Aging_{it} * Aging_{it} + \beta_3 Health_{it} + \beta_4 Pension_{it} + \beta_5 X_{it} + \varepsilon_{it} \qquad (9)$$

$$Outrent_{it} = \alpha_2 + \gamma_1 Aging_{it} + \gamma_2 Aging_{it} * Aging_{it} + \gamma_3 Health_{it} + \gamma_4 Pension_{it} + \gamma_5 X_{it} + \mu_{it} \qquad (10)$$

Equations (9) and (10) are used to identify the decisive factors of farmers' rent-in and rent-out land. $Inrent_{it}$ and $Outrent_{it}$ represent the dummy variables of rent-in and rent-out land. $Aging_{it}$ and $Aging_{it} * Aging_{it}$ represent the average age of the agricultural labor force and the square term of the average age of the agricultural labor force. $Health_{it}$ denotes the average health level of the agricultural labor force, $Pension_{it}$ indicates whether the household members have pension insurance or not, and $X_{it}$ are household characteristic variables, village characteristic variables, regional characteristic variables, and time dummy variables. $\varepsilon_{it}$ and $\mu_{it}$ represent random disturbance terms.

## 4. Analysis of Empirical Results

### 4.1. The Impact of Aging Agricultural Labor Force on Farmers' Land Transfer

Before the panel data analysis, the Hausman test is first performed to determine whether the unobserved factors involved in the model are fixed effects or random effects. The Hausman test accepts the original hypothesis condition of random effects[3]; therefore, the random effects model of panel Logit is used for estimation, and the estimation results are shown in Table 3.

Models (1) to (4) are the estimated results without and with the inclusion of control variables affecting land rent-in and rent-out. The estimation results of model (1) and model (2) show that the primary term coefficient of the aging agricultural labor force is positive, and the secondary term coefficient is negative. The estimation results of model (3) and model (4) show negative coefficients for the primary term and positive coefficients for the secondary term of the aging agricultural labor force. This indicates that, in the early stage of the aging of the agricultural labor force, when they have a certain agricultural production capacity, the aging of the agricultural labor force will promote land rent-in and inhibit land rent-out. However, with the increasing aging of the agricultural labor force, when they cannot operate their farmland, the aging of the agricultural labor force inhibits land rent-in and promotes land rent-out. Thus, Hypothesis 1 is verified.

An analysis of moderating variables shows that the deterioration of the health status of the agricultural labor force has a significant positive impact on land rent-in and rent-out. This may be because health status directly affects the working ability of the agricultural labor force, and the decline in the health status reduces the opportunities for them to engage in non-agricultural industries, thus increasing their dependence on agricultural income. When the health level of farmers is low but it does not affect their agricultural production and management capacity, they would rent-in land to achieve large-scale operation to obtain a higher agricultural income. However, when the health level of the agricultural labor force is low and affects their farming ability, they would rent-out land to maximize household income by obtaining rental income [47]. Pension insurance has a significant positive impact on land rent-out, but has no significant impact on land rent-in. This may be because farmers with pension insurance will reduce their dependence on land, thus promoting land rent-out [48].

In terms of household characteristics, education level significantly inhibits land rent-in and promotes land rent-out. The possible reason is that the higher levels of education are associated with better human capital status and people that possess them are usually more able to enter non-farm work, thereby reducing the household's dependence on land. Net agricultural income has a significant positive effect on land rent-in and a significant negative effect on land rent-out. It indicates that households with a high net agricultural income are more dependent on land and more inclined to rent-in land than to rent-out. Hired labor significantly contributes to land rent-in and discourages land rent-out. This suggests that households with hired workers have a more abundant labor and are more willing to rent-in land than to rent-out. The number of the labor force has a significant negative effect on land rent-out, indicating that the more the number of family laborers, the lower their willingness to rent out land. The per capita land area has a significant positive effect on land rent-out, indicating that the larger the per capita land area, the more willing households are to rent-out land. An analysis of the village characteristic variables shows that access to roads has a significant negative impact on land rent-in. The village type being plain has a significant positive effect on land rent-out.

**Table 3.** The impact of aging agricultural labor force on land transfer.

| Variables | Land Rent-In | | Land Rent-Out | |
|---|---|---|---|---|
| | Model (1) | Model (2) | Model (3) | Model (4) |
| Aging | 0.143 *** | 0.127 *** | −0.063 *** | −0.053 *** |
| | (0.019) | (0.019) | (0.016) | (0.017) |
| Aging $*$ Aging | −0.002 *** | −0.002 *** | 0.001 *** | 0.001 *** |
| | (0.000) | (0.000) | (0.000) | (0.000) |
| Health | | −0.106 * | | −0.176 *** |
| | | (0.060) | | (0.063) |
| Pension | | −0.011 | | 0.143 ** |
| | | (0.062) | | (0.063) |
| Eduy | | −0.042 *** | | 0.064 *** |
| | | (0.011) | | (0.011) |
| Net_agri | | 0.175 *** | | −0.177 *** |
| | | (0.015) | | (0.033) |
| Hire | | 0.848 *** | | −0.564 *** |
| | | (0.067) | | (0.084) |
| Labor | | 0.017 | | −0.061 ** |
| | | (0.025) | | (0.028) |
| Land_per | | −0.006 | | 0.010 *** |
| | | (0.005) | | (0.004) |
| Landscape | | −0.038 | | 0.207 ** |
| | | (0.086) | | (0.083) |
| Road | | −0.270 *** | | −0.081 |
| | | (0.090) | | (0.088) |
| Dum_area | control | control | control | control |
| Dum_year | control | control | control | control |
| Constant | −5.538 *** | −4.599 *** | −3.453 *** | −3.728 *** |
| | (0.453) | (0.486) | (0.429) | (0.469) |
| Wald | 260.62 | 573.33 | 665.98 | 788.82 |

Note: Robust standard errors are shown in parentheses; * $p < 0.1$, ** $p < 0.05$, *** $p < 0.01$.

### 4.2. The Effect of Aging Agricultural Labor Force on the Mechanism of Land Transfer

4.2.1. The Interaction between Aging Agricultural Labor Force and the Health Level

As farmers age, the difference in their health may become an important factor influencing land transfer. Thus, aging and health level multiplier term is constructed to reflect whether the health level constrains the impact of the aging agricultural labor force on land transfer. The estimation results are shown in Table 4.

Model (5) and model (6) report the impact of aging agricultural labor force on land rent-in without and with the addition of control variables. The results show that the coefficients of the multiplicative terms of aging agricultural labor force and health level are positive and pass the 1% significance level test. It indicates that farmers with a high health level can mitigate the negative effect of aging agricultural labor force on land rent-out relative to farmers with a low health level. In model (7) and model (8), the coefficient of the multiplicative term of aging and health level is negative and passes the 1% significance level test. This indicates that farmers with high health levels can inhibit the positive effect of aging agricultural labor force on land rent-out relative to farmers with low health levels. Thus, Hypothesis 2 was verified.

**Table 4.** The Interactive effects of aging agricultural labor force and health level on land transfer.

| Variables | Land Rent-In | | Land Rent-Out | |
|---|---|---|---|---|
| | **Model (5)** | **Model (6)** | **Model (7)** | **Model (8)** |
| Aging | −0.048 *** | −0.050 *** | 0.270 *** | 0.029 *** |
| | (0.004) | (0.005) | (0.004) | (0.004) |
| Health | −1.298 *** | −1.296 *** | 0.440 * | 0.371 |
| | (0.229) | (0.232) | (0.236) | (0.235) |
| Aging * Health | 0.025 *** | 0.025 *** | −0.012 ** | −0.011 ** |
| | (0.005) | (0.005) | (0.005) | (0.005) |
| Control variables | uncontrol | control | uncontrol | control |
| Constant | −0.677 *** | −0.244 | −5.685 *** | −5.860 *** |
| | (0.228) | (0.295) | (0.237) | (0.307) |
| Wald | 228.61 | 651.08 | 653.51 | 988.88 |

Note: Robust standard errors are shown in parentheses; * $p < 0.1$, ** $p < 0.05$, *** $p < 0.01$.

4.2.2. The Interaction between Aging Agricultural Labor Force and Pension Insurance

The old-age security function of land is an important reason for elderly farmers to rely on it. In the case of the imperfect rural old-age security system, the existence of pension insurance for farmers may lead to a different attitude towards land transfer [48]. Therefore, the multiplication item of aging and pension insurance is added to reflect whether the influence of the aging agricultural labor force on land transfer is restricted by pension insurance. The estimation results are shown in Table 5.

Models (9) to (12) report the influence of the aging agricultural labor force on land rent-in and rent-out without and with the addition of control variables. The estimated results show that the coefficients of the multiplication of aging and pension insurance are not significant, indicating that the influence of the aging agricultural labor force on land transfer is not affected by pension insurance. Hypothesis 3 has not been verified. The possible reason for this is the low level of rural pension insurance in China, where the minimum rural pension income is only 55 CNY per month in some western provinces and the maximum is only 310 CNY per month in some eastern provinces [49]. Using data from the 2018 China Health and Retirement Longitudinal Study (CHARLS), Li et al. [48] found that farmers participating in pension insurance received an average of 124.73 CNY per month, while farmers' average monthly farm income was 767.17 CNY. In addition, rental income from renting out land is still low due to the imperfect land transfer market [50]. Therefore, when pension insurance does not fully protect farmers' income, farmers may prefer to rely on land output as old-age security.

**Table 5.** The interactive effects of aging agricultural labor force and pension insurance on land transfer.

| Variables | Land Rent-In | | Land Rent-Out | |
|---|---|---|---|---|
| | **Model (9)** | **Model (10)** | **Model (11)** | **Model (12)** |
| Aging | −0.033 *** | −0.034 *** | 0.021 *** | 0.022 *** |
| | (0.003) | (0.003) | (0.003) | (0.003) |
| Pension | −0.104 | −0.088 | 0.026 | 0.033 |
| | (0.249) | (0.252) | (0.263) | (0.264) |
| Aging ∗ Pension | 0.003 | 0.003 | 0.001 | 0.001 |
| | (0.005) | (0.005) | (0.005) | (0.005) |
| Control variables | uncontrol | control | uncontrol | control |
| Constant | −1.443 *** | −1.046 *** | −5.332 *** | −5.519 *** |
| | (0.185) | (0.255) | (0.207) | (0.278) |
| Wald | 202.60 | 632.71 | 642.42 | 986.57 |

Note: Robust standard errors are shown in parentheses; *** $p < 0.01$.

### 4.3. Robustness Test

#### 4.3.1. Substitution of Independent Variables

The independent variables are replaced to test whether the results are robust. First, the average age of the agricultural labor force is replaced with the proportion of the agricultural labor force aged 60 and above in the household. Second, farmers are divided into old households, young households, and transition households according to the average age. By setting the virtual variables of old and young households and considering transition households as a baseline, the land transfer differences between old and young households are compared and analyzed [51]. Finally, the average age of the agricultural labor force is replaced with the average age of the males and females constituting the agricultural labor force in the household.

In Table 6, the estimated results shown using different measures of agricultural labor force aging are consistent with the baseline regression, which verifies the reliability of the baseline regression.

**Table 6.** Robustness tests for replacing independent variables.

| Variables | Land Rent-In | | | Land Rent-Out | | |
|---|---|---|---|---|---|---|
| | **Model (13)** | **Model (14)** | **Model (15)** | **Model (16)** | **Model (17)** | **Model (18)** |
| P_aging | −0.014 *** | | | 0.009 *** | | |
| | (0.001) | | | (0.001) | | |
| Elderly households | | −0.792 *** | | | 0.541 *** | |
| | | (0.097) | | | (0.089) | |
| Young households | | 0.333 *** | | | −0.125 * | |
| | | (0.064) | | | (0.072) | |
| Male_aging | | | −0.011 *** | | | 0.006 * |
| | | | (0.003) | | | (0.003) |
| Female_aging | | | −0.022 *** | | | 0.015 *** |
| | | | (0.003) | | | (0.003) |
| Control variables | control | control | control | control | control | control |
| Wald | 638.64 | 635.31 | 495.81 | 989.65 | 977.63 | 675.71 |

Note: Robust standard errors are shown in parentheses; * $p < 0.1$, *** $p < 0.01$.

#### 4.3.2. Test of Bivariate Probit Estimation Method

In the baseline regression, it is assumed that land rent-in and rent-out are independent. However, it has also been suggested that there may be some correlation between land rent-in and rent-out decisions [16]. For example, some farmers may rent-out land that is far from their homes and rent-in land that is closer to their homes [52]. The estimation results may be biased if the correlation between land rent-in and rent-out is not considered. In this paper, we select a bivariate model, which can consider the correlation of interference

terms between Equations (8) and (9) and carry out a joint estimation. The estimation results are shown in Table 7. In the land rent-in model, the primary term coefficient of the aging agricultural labor force is significantly positive, while the secondary term is significantly negative, which is consistent with the results of model (2), verifying the inverted U-shaped influence of the aging agricultural labor force on land rent-in. In the land rent-out model, the primary term coefficient of the aging agricultural labor force is significantly negative, while the secondary term coefficient is significantly positive, which is consistent with the regression results of model (4) and verifies the positive U-shaped influence of aging agricultural labor force on land rent-out. The results show that the regression results are still robust even considering the correlation between farmers' land rent-in and rent-out behavior.

**Table 7.** Robustness tests for replacement estimation methods.

| Variables | Land Rent-In | Land Rent-Out |
|---|---|---|
| Aging | 0.052 *** (0.007) | −0.024 *** (0.006) |
| Aging ∗ Aging | −0.001 *** (0.000) | 0.000 *** (0.000) |
| Health | −0.032 (0.022) | −0.103 *** (0.023) |
| Pension | −0.035 (0.024) | 0.069 *** (0.026) |
| Control variables | control | control |
| Constant | −1.730 *** (0.163) | −1.362 *** (0.160) |
| /athrho | −0.127 *** (0.018) | |
| Wald | 1926.11 | |

Note: Robust standard errors are shown in parentheses; *** $p < 0.01$.

## 5. Discussion

Aging is a common trend faced by human societies, and the aging process in China has been accelerating since this century. By the end of 2021, China's elderly population over 60 years of age reached 267 million, making it the only country in the world with an elderly population of more than 200 million. The level and rate of aging in rural areas are much higher than in urban areas, and the aging of the agricultural labor force poses a significant challenge to agricultural production and has received much attention from scholars [53]. Based on CFPS panel data, this study analyzes the impact of the aging agricultural labor force on land transfer and tests the moderating effects of health levels and pension insurance on land transfer.

Compared with previous studies [22,45,54], this study differs in the following aspects. First, previous studies focused on the influence of the aging agricultural labor force on agricultural production [25], agricultural technical efficiency [55], and cleaner production behavior [23], but few studies systematically analyzed the impact of the aging agricultural labor force on land transfer. For example, Wang et al. [56] only analyzed the influence of rural labor aging on land rent-out. Han et al. [22] analyzed the influence of the aging agricultural labor force on land-scale management. However, these studies only considered the linear relationship between agricultural labor aging and land transfer. With the aging of the agricultural labor force, its agricultural production capacity is a process of gradual decline. Wang et al. [57] concluded that there is an inverted U-shaped relationship between agricultural aging and technical efficiency. This study analyzed the influence of the aging of the agricultural labor force on land rent-in and rent-out by adding the squared term of the age of the agricultural labor force to the regression analysis and concluded that there was an inverted U-shaped relationship between the aging of the agricultural labor force and land rent-in and a U-shaped relationship with land rent-out. Second, previous studies have mainly analyzed the direct impact of the aging agricultural labor force on

land transfer [21,58], this study incorporated health level and pension insurance into a unified theoretical analysis framework and expanded the research on the mechanism of agricultural labor aging on land transfer. Consistent with the study of Wang et al. [56], this study found that, compared with farmers with low health levels, farmers with high health levels could alleviate the negative impact of the aging agricultural labor force on land rent-in and restrain the positive impact of the aging agricultural labor force on land rent-out. Compared with existing studies, for example, Hu et al. [59] found that the enrolment in the public pension system increases the scale of farmland transfer. Wang et al. [36] found that farmers' participation in the new rural pension system increases the probability of land transfers. This study found that the endowment insurance did not play a significant moderating role in the impact of the aging agricultural labor force on land rent-in and rent-out. The possible reason is that pension insurance provides a formal institutional guarantee for the old-age security of farmers; the current level of pension insurance, however, is relatively low, and it is difficult to meet the basic needs of farmers [49]. Li et al. [48] found that the average monthly income of farmers with pension insurance was 124.73 CNY in 2018, but the average monthly consumption expenditure was 1010.33 CNY. If farmers cannot obtain enough income from the pension insurance to offset the loss of land, land will continue to be a basic old-age security mechanism for farmers, thus inhibiting the land transfer of elderly farmers. Meanwhile, China's rural land transfer market-oriented operation mechanism has not been fully established; there is no unified and standardized land transfer market [50]. Most of the land transfer is spontaneous transfer between farmers; hence, the transfer price is simply negotiated by both sides in advance. The land transfer price is too low, which cannot guarantee a normal income to rural families. Therefore, pension insurance does not significantly regulate the impact of the aging agricultural labor force on land transfer.

## 6. Conclusions and Implications

Aging is a major trend in the evolution of China's rural population structure. Using CFPS survey data, this study empirically analyzed the impact of the aging agricultural labor force on land transfer and its mechanisms. The main conclusions are as follows: The aging of the agricultural labor force has an inverted U-shaped impact on land rent-in and a U-shaped impact on land rent-out. Thus, in the early stage, the aging of the agricultural labor force promotes land rent-in and inhibits land rent-out. However, as aging advances, the aging of the agricultural labor force inhibits land rent-in and promotes land rent-out. The mechanism analysis shows that the impact of aging on land transfer is governed by the health levels of the labor force; agricultural labor forces with high health levels can mitigate the negative impact of the aging agricultural labor force on land rent-in and inhibit the positive impact of the aging agricultural labor force on land rent-out compared to agricultural labor forces with low health levels. In contrast, pension insurance does not play a significant moderating role in the impact of the aging agricultural labor on land rent-in and rent-out.

Based on the above research findings, the following policy recommendations are proposed. First, the construction of the rural land transfer market needs to be strengthened. The government should enhance the management of contracts of land management rights transfer, promote the standardized and orderly transfer of rural land management rights, and guide farmers to carry out the orderly transfer of land management rights following the market mechanism. Second, the treatment level of rural pensions needs to be improved. Investment in rural social security funds needs to be increased, a pension insurance system that is in line with the level of economic development need to be established, and the treatment standard of rural pension insurance needs to be gradually raised. This will weaken the old-age security function of land and reduce aging farmers' dependence on it.

In addition, this study has some limitations. First, the average age of the agricultural labor force is used to measure aging. However, aging is a dynamic concept. With the improvement of nutrition levels and medical and health conditions, a 60-year-old farmer

in the future may be more productive than a 50-year-old farmer now. At the same time, there is no retirement age for the labor force in rural areas, and most people are still engaged in agricultural production even after turning 60 years old. Second, due to data limitations, this study only analyzes the impact of an aging agricultural labor force on land transfer behavior, but not its impact on the land transfer area. Future studies can examine the impact of the aging agricultural labor force on the land transfer area.

**Author Contributions:** Conceptualization, C.L.; data curation, J.W. and X.L.; formal analysis, X.L. and T.F.; writing—original draft, C.L. and J.W.; writing—review and editing, C.L. All authors have read and agreed to the published version of the manuscript.

**Funding:** Soft Science Project of Zhejiang Provincial Department of Science and Technology (NO. 2022C35066) and Doctor Training Program of Jiyang College, Zhejiang Agriculture and Forestry University (RC2022D03).

**Data Availability Statement:** The data presented in this study are available on request from corresponding author.

**Acknowledgments:** We thank the editors and anonymous referees for their helpful comments and suggestions.

**Conflicts of Interest:** The authors declare no conflict of interest.

## Notes

[1] Among the 26,909 samples used in this study, 7313 households were included, of which 2845 were tracked five times, 1441 were tracked four times, and 1561 were tracked three times. Therefore, sample households with a tracking rate of three or more accounted for 79.95% of the total sample.

[2] The agricultural labor force was categorized as very healthy, relatively healthy, and healthy in terms of self-rated health levels as high health levels, and fair and unhealthy in terms of self-rated health levels as low health levels.

[3] In the land rent-in and land rent-out models, the Hausman test results in $p$-values equal to 0.718 and 0.475, respectively, both of which are much greater than 0.05; hence, the random effects model is used for the regression.

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
