# Peer review of "Impacts of Aging Agricultural Labor Force on Land Transfer: An Empirical Analysis Based on the China Family Panel Studies"

_land, doi:10.3390/land12020295_

Round 1
Reviewer 1 Report (Previous Reviewer 1)
The authors took into account the reviewer comments and recommendations
Author Response
On behalf of my co-authors,Thank you for your encouragement and recognition of the paper!
Reviewer 2 Report (Previous Reviewer 2)
The manuscript has improved, however, some of the points I raised in my review have been answered relatively briefly.
Author Response
On behalf of my co-authors,Thank you for your encouragement and recognition of the paper!
Reviewer 3 Report (Previous Reviewer 3)
This study analyzed the impact of aging agricultural labor on land transfer and its mechanism. Overall, this study addresses a topic of high relevance for research and also for practice. However, I believe some issues need revision and clarification.

Author Response
On behalf of my co-authors, we thank you very much for giving us an opportunity to revise our manuscript. We thank the reviewer for these insightful and useful suggestions. Point-to-point responses to these comments are provided below. Corresponding changes have been made in the paper.

This manuscript is a resubmission of an earlier submission. The following is a list of the peer review reports and author responses from that submission.
Round 1
Reviewer 1 Report
The manuscript is significant for the journal’s profile, the relationship between Rural Aging and Land Transfer is a subject that could bring some new information and has some practical solutions.
However, the paper is not fit to be published in its current form. There is need for revision and improvements in order to be published.
Introduction
I suggest that the Introduction chap. could be also developed based also on some international literature, projects results, some official reports having in view not almost Chinese references.
Please extend also the research question which is not well developed.
Results
The results are very well presented and described.
Discussions
I suggest to add a section dedicated to uncertainties and limitations of the study.
Conclusions
In their current form, the Conclusions look like a very short summary.
The connection between results and some research projects could also be approached.
What can your results be used for?
The manuscript is based almost exclusively on Chinese reference list.
Author Response
Dear reviewer/professor:
We thank the reviewer for these insightful and useful suggestions. Please refer to the attachment for details.
Best regards,
Sincerely, yours
Chaozhu Li

Reviewer 2 Report
Basically, I like the research questions formulated in the manuscript, which I consider relevant.
The main problem I see with the manuscript is that I suppose the authors are processing their research from a relatively strong “inland” perspective. Of course, there would be nothing wrong with that if they wanted to publish the manuscript in a Chinese-scoped journal published in the People's Republic of China. However, since Land is a global-scoped journal for global readers in my interpretation, it should be even better written for global readers (otherwise they may not have adequate information about some of the factors mentioned in the manuscript).
The article's title says "Impacts of Rural Aging on Land Transfer". At the same time, the authors do not explain (it would suffice briefly) what they consider to be rural or urban spaces and settlements in the People's Republic of China in terms of their research questions (I explain this aspect better later in my review).
The terms "land Inflow" and "land Outflow" are mentioned in several places in the manuscript. However, I do not think these key terms are clearly defined by the authors (these concepts are less common in works in international literature published by non-Chinese researchers anyway), which I think would be appropriate to do right at the beginning of their study. And I suppose it would be better to use synonyms of these in the manuscript, as well, such synonyms which are more common in similar literature published outside of China.
The authors perform complex statistical data analysis. However, little information is provided about the CFPS on which the essential statistical data of the manuscript are based. In this connection, I have some questions that I consider necessary to clarify in the manuscript.
To what extent is CFPS considered representative of its research questions, in relation to the variables studied by the authors, for the People's Republic of China as a whole?
How proper do you think of using 2010 or 2012 data CFPS in 2022?
What is the reason for the latest year of CFPS being 2016?
The authors analyze a wide variety of variable types (e.g., absolute value, % ratio, intensity ratio). However, they should write a little more about this, and whether the many types of variables posed some kind of methodological-counting challenge for them.
I assume it is not entirely clear in the current version of the manuscript exactly what the authors' research is about spatially within China. For their study, do they define all the People's Republic of China rural areas as rural? It would be nice to define the exact area of their study somehow more precisely, and possibly also display them on a map, at least schematically.
If the authors analyze all the rural areas of the People's Republic of China together and jointly, then the question arises as to whether it would not have been at least tangentially reasonable to examine the variables included in the research in relation to e.g. 2-3 groups of provinces and to check how the geographical location (e.g., eastern or western groups of provinces) can cause differences statistically. Indeed, given the extended population and area of the People's Republic of China, it seems to me a bit static and geographically indifferent to analyze the entire study area in one (differences in territorial, e.g. province groups, can be analyzed). There may be at least some variables that may differ spatially. However, there are still references in the study to spatiality and the role of geography (e.g., Table 2, p. 4). These are useful but given the population and territorial dimensions of the People's Republic of China, I do not think they are sufficient.
Part 4.1.1 (Omitted variable test) is useful and shows, for example, the role of family policy. At the same time, I wonder how the authors define "(2) whether the village is in a region where the one-child policy is strictly implemented", so how can “strictly” be defined? In connection with this, I wonder how justified it would be to look at the proportion of the non-Chinese population as a variable in this section (if there is any data for this). According to my information, non-Chinese minorities may have had two children in the one-child policy era, which may have some impact on provincial-level demographics in certain provinces even now.
The authors' policy proposals should be appreciated (p. 14.). However, the first one is not entirely clear to me ("(1) accelerating the professional farmer cultivation plan, encouraging young labor to engage in agricultural-related jobs"). Do the authors mean local rural youth here (and not possibly migrated ones from urban areas)? I do not have enough information about the conditions in China in this regard, but globally I am afraid that it can be concluded that this is a very difficult challenge and it is not easy, for example, due to the growing urbanization and increasing working condition expectations, to engage young people in agricultural-related jobs. Instead, given the manuscript's statistical findings (e.g., demographics), perhaps the intensification of agricultural production, e.g., the amount of capital invested per worker or unit of land, could be increased if there is sense to advise this.
Author Response

(The authors gave the same response as above.)

Reviewer 3 Report
The manuscript has studied the impact of rural
aging on land transfer. This study used data from the Chinese family panel
studies from 2010-2016 and implemented Probit and Tobit models to explore the
impacts of rural aging on land transfer. The results show that rural aging has
a significant negative effect on land input rate and land input level. Overall,
this study addresses a topic of high relevance for research and also for
practice. However, I believe some issues need revision and clarification.

Author Response

(The authors gave the same response as above.)

Round 2
Reviewer 1 Report
The authors addressed all the comments, the paper could be published in revised form
Author Response
Thank you for your appreciation of our articles!
Reviewer 2 Report
Your manuscript has been improved a lot. However, the content of the authors' answers below should be incorporated better and somewhat more extensively even in the improved manuscript, I assume, because these pieces of information would make the research and its methodological limitations even more clearer.
Comment 4. The authors perform complex statistical data analysis. However, little information is provided about the CFPS on which the essential statistical data of the manuscript are based. In this connection, I have some questions that I consider necessary to clarify in the manuscript. To what extent is CFPS considered representative of its research questions, in relation to the variables studied by the authors, for the People's Republic of China as a whole?
Response 4. Thank you for your suggestion! The modification is described as follows: The China Family Panel Studies (CFPS), launched by Peking University, is a nearly nationwide, comprehensive, longitudinal social survey that is intended to serve research needs on a large variety of social phenomena in contemporary China(Xie and Hu, 2014).The survey sample covers 25 provinces or administrative (excluding Xinjiang, Tibet, Qinghai, Inner Mongolia, Ningxia , Hainan, Hong Kong, Macao and Taiwan) equivalents represent 95 percent of the total population in China (excluding Hong Kong, Macao and Taiwan). Therefore CFPS is a nearly nationally representative. The distribution of sample provinces is shown in Figure 1.
Xie Y, Hu J. An introduction to the China family panel studies (CFPS)[J]. Chinese sociological review, 2014, 47(1): 3-29.
In Xie and Lu’s article in 2015, They explained why CFPS is nationally representative, The details are as follows: While true national representativeness would be ideal, the CFPS project did not have the resources to conduct longitudinal surveys in remote, minority regions, especially where traveling would be very difficult, and non-Han languages would likely be required.
I did not even know that travel and translation costs are so high for these regions, e.g. even in the case of Inner Mongolia which is a region located relatively closer to Central China.
To contain costs, a decision was made not to include Xinjiang, Tibet, Qinghai, Inner Mongolia, Ningxia and Hainan. Needless to say, Hong Kong, Macao and Taiwan were also excluded.
Ok, but in this sense, because CFPS project is not completely representative of non-Han regions in China, I am afraid that the title and the text of body of your manuscript may not involve these geographical limitation well (e. g. when you refer to 'China' as a whole).
However, the remaining 25 provinces or administrative equivalents represent 94.5 percent of the total population in China (excluding Hong Kong, Macao and Taiwan). Thus, we state that the CFPS sample is ‘nearly nationally representative’ or, for convenience, simply ‘nationally representative’.
Comment 5. How proper do you think of using 2010 or 2012 data CFPS in 2022?
Response 5. Thank you for your suggestion! Our answer is as follows: Due to the limitation of data, this study only used the data of 2010 and 2012 when analyzed the impact of rural population aging on the area of land transfer. CFPS only investigated the area of farmer' land transfer in 2010 and 2012. Therefore, this study used the data of 2010 and 2012. This is also a deficiency of our research.
Comment 6. What is the reason for the latest year of CFPS being 2016?
Response 6. Thank you for your suggestion! Your suggestion reminded us. We went to CFPS's official website to apply for the latest data. At present, the latest survey data of CFPS is 2020, but at present, CFPS only released the individual database data of 2020. This study uses data from three levels: family, individual and community. Therefore, this study updates the research data to 2018. Descriptive analysis and empirical analysis were conducted again, and the regression results did not change significantly compared with the previous one.
Author Response
Thank you for your suggestions. The authors have carefully revised the article according to your suggestions. Please see the attachment for the revision instructions.

Reviewer 3 Report
Most of my remarks from the previous round of revision have been addressed. Yet, there are some issues need to be addressed.

Author Response

(The authors gave the same response as above.)
